# Added Value of Cone-Beam Computed Tomography for Detecting Hepatocellular Carcinomas and Feeding Arteries during Transcatheter Arterial Chemoembolization Focusing on Radiation Exposure

**DOI:** 10.3390/medicina59061121

**Published:** 2023-06-11

**Authors:** Duk-Ju Kim, In Chul-Nam, Sung-Eun Park, Doo-Ri Kim, Jeong-Sub Lee, Bong-Soo Kim, Guk-Myung Choi, JeongJae Kim, Jung-Ho Won

**Affiliations:** 1Department of Radiology, Jeju National University, School of Medicine, Jeju Natuional University Hospital, 15, Aran 13-gil, Jeju 63241, Republic of Korea; dongguri518@gmail.com (D.-J.K.); antisors@naver.com (D.-R.K.); shinshlee@naver.com (J.-S.L.); 671228kbs@naver.com (B.-S.K.); choigm@jejunu.ac.kr (G.-M.C.); jsquare8057@gmail.com (J.K.); 2Department of Radiology, Gyeongsang National University, School of Medicine, Gyeongsang National University Changwon Hospital, 11 Samjeongja-ro, Seongsan-gu, Changwon 51472, Republic of Korea; uneyes@gnuh.co.kr; 3Department of Radiology, Gyeongsang National University, School of Medicine, Gyeongsang National University Hospital, Jinju 52727, Republic of Korea; circlehoya@naver.com

**Keywords:** chemoembolization, hepatocellular carcinoma, cone-beam computed tomography, radiation exposure, sensitivity

## Abstract

*Background and Objectives*: This study aimed to evaluate the added value of cone-beam computed tomography (CBCT) for detecting hepatocellular carcinomas (HCC) and feeding arteries during transcatheter arterial chemoembolization (TACE). *Material and methods*: Seventy-six patients underwent TACE and CBCT. We subcategorized patients into groups I (61 patients: possible superselection of tumor/feeding arteries) and II (15 patients: limited superselection of tumor/feeding arteries). We evaluated fluoroscopy time and radiation dose during TACE. Two blinded radiologists independently performed an interval reading based on digital subtraction angiography (DSA) imaging only and DSA combined with CBCT in group I. *Result*: The mean total fluoroscopy time was 1456.3 ± 605.6 s. The mean dose–area product (DAP), mean DAP of CBCT, and mean ratio of DAP of CBCT to total DAP was 137.1 ± 69.2 Gy cm^2^, 18.3 ± 7.1 Gy cm^2^, and 13.3%, respectively. The sensitivity for detecting HCC increased after the additional CBCT reading, from 69.6% to 97.3% and 69.6% to 96.4% for readers 1 and 2, respectively. The sensitivity for detecting feeding arteries increased from 60.3% to 96.6% and 63.8% to 97.4% for readers 1 and 2, respectively. *Conclusions*: CBCT can increase sensitivity for detecting HCCs and feeding arteries without significantly increasing the radiation exposure.

## 1. Introduction

Hepatocellular carcinoma (HCC) is the most common primary liver cancer, accounting for 70% to 90% cases, and it is the sixth most commonly diagnosed and fourth most fatal cancer worldwide [1,2,3]. The Barcelona Clinic Liver Cancer staging system, which considers the tumor stage, liver function, and performance status, can be used to help recommend the treatment strategy according to each stage [4,5]. Transcatheter arterial chemoembolization (TACE) is the first-line treatment option for patients with the intermediate stage of HCC [6,7,8,9]. TACE can be classified into two types: conventional TACE (C-TACE) and TACE with drug-eluting beads (DEB-TACE). C-TACE was the first established option to treat the intermediate stage of HCC and works by delivering a cytotoxic chemotherapeutic agent saturated with ethiodized oil (Lipiodol, Guerbet, Villepinte) and occluding tumor-feeding arteries with an embolic agent to maximize cytotoxic and ischemic effects [6,7]. Although C-TACE has been reported to be associated with survival gain in patients with HCC, it has also been associated with a high incidence of systemic toxicity induced by the rapid circulation of chemotherapeutic-agent-loaded lipiodol [10,11]. Additionally, cTACE presents numerous challenges due to the absence of standardization at every step [12]. This deficiency in standardization introduces a wide range of biases that hinder the comparison of previously published studies on patients who underwent cTACE treatment. In a recent work, Renzulli et al. [13] evaluated the in vitro chemical and physical characteristics and behaviors over time of emulsions for cTACE prepared using two different methods and assessed the intra- and inter-operator variabilities in the preparation processes. In this interesting study, the mean droplet diameter decreased non-significantly when the number of pumping exchanges increased and increased significantly over time for both water-in-oil and chemotherapeutic-in-oil techniques. The droplets returned to their initial diameters after re-mixing and there were no significant differences in the intra- and inter-operator variabilities. According to their investigation, any interventional radiologists, regardless of their experience, may prepare these emulsions and these data may represent a set of instructions to standardize cTACE. A high incidence of systemic toxicity and a lack of standardization of cTACE led to the development of DEB-TACE, which uses microspheres as drug carriers to slowly release chemotherapeutic agents and substantially reduce the concentration of systemic chemotherapeutic agents [6,14,15]. In addition, balloon-occluded TACE (B-TACE) was recently developed and aims to promote the dense concentration of chemotherapeutic agents and lipiodol in tumors. Under the occlusion of feeding arteries using a microballoon catheter, an emulsion of chemotherapeutic agents with lipidol was injected followed by embolic materials [16,17].

TACE has been traditionally performed with conventional digital subtraction angiography (DSA) imaging, and it has often been challenging to visualize the tumor and feeding arteries. It has been reported that the detection rate of HCC and the feeding arteries can be increased considerably by performing cone-beam computed tomography (CBCT) during TACE, which can provide intraprocedural three-dimensional volumetric imaging during TACE [18,19]. Vania et al. [18] suggested that CBCT is becoming an essential tool during TACE for localizing tumors and feeding arteries, planning and guiding for catheterization, and for the intraprocedural evaluation of treatment success. Schernthaner et al. [20] also reported that only 45.9% of liver cancers were detected using DSA, whereas delayed-phase CBCT showed a significantly higher detectability rate of up to 93.4%. Although CBCT can be useful during TACE, it can increase the radiation exposure and fluoroscopy time for patients and operators. Jonczyk et al. [21] reported that CBCT increases the radiation exposure in TACE. However, the capability of CBCT to detect HCC with feeding arteries and overlay in real time during fluoroscopy helps to facilitate TACE with a resultant reduction in the dose–area production (DAP) by up to 46%.

Patients with HCC often require repeated TACE according to the viability or recurrence of the tumor. Therefore, the radiation dose can be an especially important issue for both the operator and patients.

Therefore, our study aimed to evaluate the added value of CBCT for detecting HCC and feeding arteries during TACE, focusing on radiation exposure.

## 2. Materials and Methods

### 2.1. Patients

The Institutional Review Board of Gyeongsang National University Changwon Hospital approved the current study (No.: GNUCH 2022-02-010). Owing to the retrospective nature of the study, no patient approval or informed consent was required.

We identified 85 patients who underwent TACE between 1 January 2020 and 30 June 2021 in our institutional database. We excluded 8 patients that did not have a radiation dose report or were lost to follow-up. We reviewed each patient’s medical history, including age, sex, Child–Pugh score, performance status, and reason for TACE. Every enrolled patient underwent contrast-enhanced liver CT or magnetic resonance imaging (MRI) within 54 days (mean: 18 days, range: 1–54 days). We reviewed the CT or MRI images before the procedure, and the patients were subsequently divided into two groups according to the multiplicity of HCCs. The groups were defined as follows: group I—one or several HCCs and possible superselection of tumor/feeding arteries—and group II—multiple or diffuse HCC, multiple feeding arteries, multiple extrahepatic feeding arteries, and impossible superselection of tumor/feeding arteries. We evaluated the technical success and tumor response to TACE according to the modified Response Evaluation Criteria in Solid Tumors (mRECIST) criteria after 1 month [22], which were defined as follows: technical success—successful achievement of lipiodol uptake in the target lesion and tumor response to TACE according to mRECIST; complete response (CR)—no intratumoral arterial enhancement in all target lesions; partial response (PR)—≥30% reduction in the sum of diameters of viable (enhancement in the arterial phase) target lesions; stable disease—features classifiable as neither PR nor progressive disease (PD); and PD—≥20% increase in the sum of the diameters of the viable target lesions. We also evaluated the procedural data, including the total fluoroscopy time, total radiation dose, and radiation dose of CBCT. Short-term procedure-related complication rates and management were recorded according to the Cardiovascular and Interventional Radiological Society of Europe standards of practice on TACE [23]. Post-embolization syndrome (PES) was not considered a complication because PES is usually self-limiting and usually lasts only a few days within the first 2 weeks after the procedure [23]. We divided complications into minor and major (all events, such as hepatobiliary injury, hemorrhage, and vascular damage that need some kind of intervention). Minor complications were defined as adverse events that did not need any or needed only symptomatic treatment, while major complications were those that necessitated a wide range of radiological, medical, and surgical procedures. The case accrual process is summarized in Figure 1.

### 2.2. TACE Procedure Technique

Two interventional radiologists with 2 and 5 years of experience in interventional radiology performed all procedures in an angio-suite (AlluraClarity FD20 with XperCT Roll protocol; Philips Healthcare, Andover, MA, USA).

First, under ultrasound (USG) guidance, local anesthesia was injected above the common femoral artery. Subsequently, common femoral artery access was accomplished using the Sheldinger technique. Celiac arteriography was performed with a 5-Fr guiding catheter. Additional CBCT was performed after placing a 5-Fr catheter in the proper hepatic artery or common hepatic artery. If necessary, we performed additional DSA for route mapping according to the variants of hepatic artery anatomy, tumor, and feeding arteries, such as the right inferior phrenic artery, internal thoracic artery, or aberrant hepatic artery arising from the superior mesenteric artery. Most of the DSAs were performed with three images per second with contrast injected by an injector. The contrast amount and flow rates were adjusted to the anatomy; usually, 15 mL to 20 mL of contrast agent with an injection rate of 3 mL/s to 4 mL/s was injected in the hepatic artery proper. After selective microcatheterization was achieved in the feeding artery, C-TACE was performed using an emulsion of ethiodized oil (Lipiodol, Guerbet, Princeton, NJ, USA) mixture with 30 mg to 50 mg of doxorubicin followed by gelatin sponge embolization. Alternatively, DEB-TACE was performed using 100 um Embozene TANDEM (Microspeheres for Embolization; Boston Scientific, Marlborough, MA, USA), or B-TACE was performed using a microballoon catheter. The endpoint of a chemotherapeutic agent was determined by visualization of the perilesional portal vein, and the endpoint of embolization was determined by stasis of the arterial flow. A final DSA was performed to confirm the occlusion of the feeding arteries and residual tumor blush. In case of uncertain coverage of the tumor with lipiodol, an additional CBCT without contrast injection was performed.

According to the “as low as reasonably achievable” principle, we attempted to reduce the X-ray dose as much as possible during the procedure by using high tube voltage, low tube current, beam collimation, increasing the distance between the patient and the beam source, having an intermittent beam time, and avoiding magnification.

### 2.3. CBCT Protocols

All CBCT images were acquired during the procedure with a single bolus contrast injection by contrast injector after placing a 5-Fr catheter in the proper hepatic artery or microcatheter in the potentially proximal portion of the feeding artery (Figure 2). The precise parameters were total scanning angle, 240 degrees; rotation speed, 40.85 degrees/s; and scan time, 5.2 s. The raw image data were transferred to an external workstation (EmboGuide, Philips Healthcare), where images were reconstructed to produce a 3-dimensional vessel roadmap.

### 2.4. Assessment of the Tumor and Feeding Artery: Image Analyses

Two radiologists with 10 and 13 years of experience in evaluating liver CT and MR who did not participate in the TACE procedures independently reviewed the DSA and CBCT images. Both radiologists were blinded to all clinical information except for the information that patients were at risk for HCC.

At the first session, each reader reviewed the DSA images from patients in group I. Each reader recorded whether the tumor or feeding arteries were detected according to a 5-point confidence scoring system: 1: definitely no detectable HCC or feeding arteries; 2: probably no detectable HCC or feeding arteries; 3: possibly detectable HCC or feeding arteries; 4: probably detectable HCC or feeding arteries; and 5: definitely detectable HCC and feeding arteries. One month later, each reader scored their confidence level for detecting HCC and feeding arteries with a combined image set of CBCT and DSA images using the same scoring system. Liver CT or MR before the TACE and the follow-up unenhanced or enhanced CT served as the diagnostic reference standards for HCC and the associated feeding arteries.

### 2.5. Statistical Analysis

An independent *t*-test and McNemar test were used, and a *p*-value of <0.05 was considered statistically significant. Continuous variables were expressed as the mean ± standard deviation.

Sensitivity was calculated on the assumption that a confidence level of three or higher was positive for detecting HCC or feeding arteries.

The interobserver agreement between the two readers was calculated using weighted kappa statistics. The weighted kappa value was interpreted as follows: 0, no agreement; 0.01 to 0.20, slight agreement; 0.21 to 0.40, fair agreement; 0.41 to 0.60, moderate agreement; 0.61 to 0.80, substantial agreement; and 0.81 to 1.00, almost perfect agreement.

## 3. Results

A total of 84 consecutive patients underwent TACE. Three patients were excluded due to a lack of radiation reports, and five were excluded due to being lost to follow-up. Finally, a total of 76 patients were enrolled in this study. There were 66 (86.8%) men and 10 (13.2%) women (age range, 52–82 years; mean age, 65.2 ± 9.3 years) included. All the patients were in the intermediate stage according to the Barcelona Clinic Liver Cancer staging system. Of the 76 patients included, 49 (64.5%), 17 (22.4%), 9 (11.8%), and 1 (1.3%) had a Child–Pugh score of A5, A6, B7, and B8, respectively. All patients had a grade 0 for performance status according to the Eastern Cooperative Oncology Group scoring system. Seventy-one patients underwent cTACE, two underwent DEB-TACE, and the remaining three underwent B-TACE.

The technical success, defined as the successful achievement of complete lipiodol uptake of the target lesion, was 100%. In terms of the tumor response to TACE according to mRECIST, 42 patients (55.3%) had a CR, 25 patients (32.9%) had a PR, 2 patients (2.6%) had stable disease, and 7 patients had PD (9.2%). The mean follow-up interval was 26 ± 11 days.

The mean total fluoroscopy time was 1456.3 s ± 605.6 (range: 627–3519). We used the dose–area product (DAP) reported by the angiography machine to assess the X-ray dose. The mean total DAP was 137.1 ± 69.2 Gy cm^2^ (range: 53.1–380.1). In addition, the mean DAP of CBCT was 18.3 ± 7.1 Gy cm^2^ (range: 7.7–48.3), and the mean ratio of CBCT to total DAP was 13.3%. Of the 76 patients, 61 and 15 were included in groups I and II, respectively. The mean fluoroscopy time was 1366.2 ± 524.4 s (range: 627–3231) in group I and 1822.7 ± 779.1 s (range: 925–3519) in group II (*p* = 0.046). The mean DAP was 128.1 ± 66.7 Gy cm^2^ (range: 53.1–380.1) in group I and 174 ± 69.2 Gy cm^2^ (range: 67.7–250.1) in group II (*p* = 0.02). The mean ratio of CBCT to total DAP was 14.3% in group I and 10.2% in group II. The results are summarized in Table 1. Box and whisker plots compare the fluoroscopy time and DAP between groups I and II (Figure 3 and Figure 4). A pie chart compares the ratio of CBCT to total DAP between the two groups (Figure 5).

On assessing the detection rate of tumors and feeding arteries in group I by both readers, the kappa values for detecting tumors were 0.958 (*p* < 0.001) and 0.853 (*p* < 0.001) for DSA alone and combined CBCT and DSA, respectively. The kappa values of DSA alone and combined CBCT and DSA for detecting feeding arteries were 0.890 (*p* < 0.001) and 0.853 (*p* < 0.001), respectively, suggesting almost perfect concordant results. There were 112 target tumors and 116 feeding arteries in the 61 group I patients. The sensitivity of DSA alone for detecting tumors and feeding arteries was 69.6% and 60.3% for reader 1 and 69.6% and 63.8% for reader 2. After the additional reading of CBCT, the sensitivity for detecting tumors and the feeding arteries increased, from 69.6% to 97.3% and 60.3% to 96.6% for reader 1, and from 69.6% to 96.4% and 63.8% to 97.4% for reader 2, respectively, after combined reading of CBCT and DSA (Figure 6). The results are summarized in Table 2.

PES occurred in 28 patients (36.8%) and was characterized by nausea, vomiting, fever, pain, and fatigue, necessitating no or only symptomatic treatment. Complications occurred in three patients (3.9%), one patient with liver infarction needed a prolonged hospital stay and medical treatment, one patient with a small liver abscess was treated with antibiotics, and the remaining patient with acute pancreatitis required conservative treatment.

## 4. Discussion

Since the first report of the selective deposition of lipiodol in HCC when injected into the hepatic artery in 1983 [24], TACE has been considered the standard treatment for patients with the intermediate stage of HCC, according to the Barcelona Clinic Liver Cancer system. Although CR was achieved with only a single treatment in some cases, in many cases it needed additional TACE to treat a viable portion of HCC or the progression of HCC. In addition, most patients with HCC have a high risk of recurrence or occurrence of HCC owing to underlying liver cirrhosis or chronic liver disease [2,3,9] and, therefore, often require further treatment. As such, TACE should be understood as a series of sessions rather than a one-time treatment. Therefore, radiation exposure during the procedure is particularly important for patients and the operator.

In our study, TACE with CBCT was found to be effective and safe for the treatment of HCC, with a technical success rate of 100%; 67 patients (88.2%) had CR or PR, and the major complication rate was 3.9%. PES was the most frequent adverse effect associated with TACE, accounting for 36.8%, and was characterized by abdominal pain and fever followed by transiently elevated liver enzyme levels, which were self-limiting. These results were similar to those of several recent studies. Lencioni et al. [25] reported in their systemic review that a transient increase in liver enzymes was the most commonly observed adverse effect (52% of patients who underwent TACE), followed by PES (47.7%). Quinto et al. [26] reported that PES occurred in 22% of patients and major complications occurred in 5%. In our study, most of the adverse effects were PES (36.8%), and only 3.9% were major complications that required additional treatment and a prolonged hospital stay. According to the Cardiovascular and Interventional Radiological Society of Europe standards of practice on TACE guidelines [23], our complication rate was acceptable.

We found that the sensitivity for detecting HCC and feeding arteries significantly increased by approximately 20–35% after adding CBCT to DSA. In addition, the mean confidence level of both readers for detecting HCC and feeding arteries by CBCT was significantly higher than that on DSA alone. These results are concordant with previous studies. A recent systematic review and meta-analysis [27] showed that the pooled sensitivities of CBCT for detecting tumors and feeding arteries were 90% and 93%, respectively. In contrast, the pooled sensitivities of DSA for tumor detection and feeding arteries were 67% and 55%, respectively. Lucatelli et al. [28] also reported a similar result, detailing that dual-phase CBCT had a better diagnostic performance than pre-examination CT or MR in detecting and characterizing HCC during TACE with high sensitivity and accuracy. Therefore, adding CBCT during the TACE procedure is definitely helpful in finding the tumor and its feeding vessel.

The greatly Increased detection rate of tumors and feeding arteries when adding CBCT to DSA is quite encouraging. With recent advancements in imaging techniques, surveillance programs for HCC detection in chronic liver disease are being implemented in Europe, some Asian countries, and the United States, and future strategies to improve these surveillance programs are being proposed. According to a meta-analysis by Tzartzeva et al. [29], while USG alone has 84% sensitivity for HCC of any stage, it only has 45% sensitivity for early stage HCC, detecting only 4 out of 10 patients. Their meta-analysis confirmed that the sensitivity of HCC detection increases to 63% when USG is combined with alpha-fetoprotein (AFP) measurement, and they recommend adding AFP to USG for surveillance in clinical practice. Furthermore, Park et al. [30] reported that USG showed 27.9% sensitivity and non-enhanced MRI showed 79.1% sensitivity for HCC surveillance in high-risk patients, suggesting that non-enhanced MRI could be a replacement modality for USG. A recent meta-analysis by Lee et al. [31] found that MRI had higher per-lesion sensitivity than CT (80% vs. 68%, *p* = 0.0023) for evaluating HCC. Moreover, ethoxybenzyl-diethylenetriamine pentaacetic acid (EOB)-MRI showed significantly higher per-lesion sensitivity than MRI performed with other contrast agents (87% vs. 74%, *p* = 0.03) [31]. Accordingly, EOB-MRI should be the preferred imaging modality for the diagnosis of HCC in patients with chronic liver disease. Based on these results, a new diagnostic algorithm for hepatocellular carcinoma using EOB-MRI, based on Japanese guidelines but adapted to the Western world for patients under surveillance for chronic liver disease, has been proposed in Europe [32]. In contrast to the sensitivity of 27.9–45% for USG alone, 63% USG with AFP, 46% for CT, 80% for MRI, and 80% for EOB-MRI shown in literature reviews [29,30,31,32], in our study, the sensitivity exceeded 95% when DSA and CBCT were interpreted together. This suggests that CBCT could be considered as part of future strategies for HCC surveillance when HCC cannot be confirmed based on imaging alone. However, further studies are needed to determine whether DSA, as an invasive technique, is superior to other invasive methods such as biopsy or noninvasive imaging modalities. Furthermore, our study may have a selection bias as we performed CBCT only on patients who were already diagnosed with HCC through surveillance. Further research, such as randomized clinical trials, is needed to address this issue.

However, the use of CBCT inevitably results in additional radiation exposure, despite the benefit of adding CBCT to TACE. The mean dose of radiation of added CBCT was 18.3 ± 7.1 Gy cm^2^, and the mean ratio of DAP of CBCT to total DAP was 13.3%. Compared with a recent study in which the results showed that additional CBCT increased DAP by 6% [21], our additional radiation dose of CBCT seems to be high. However, in their study, the total mean DAP was 318.9 Gy cm^2^, which was 2.3 times higher than ours. Furthermore, the overall mean DAP of CBCT was 17.9 Gy cm^2^ in their study, which was similar to the current study. Therefore, a ratio of 10 to 15% of CBCT to the total radiation dose in our study seems to be acceptable. Moreover, in our study, the total radiation dose of group II was significantly higher than group I, whereas the ratio of the radiation dose of CBCT to the total radiation dose was lower in group II than in group I. This suggests that as the total radiation dose increases, the ratio of the radiation dose of CBCT to the total radiation dose decreases. Based on these findings, even if the ratio of the radiation dose of CBCT to the total radiation dose were to increase, the total radiation dose would decrease, so this also seems acceptable.

Compared with the data reported in the literature, the total mean radiation dose in this study seems low. Schegerer et al. [33] reported that the 75th percentile of the median value of DAP of TACE was 241 Gy cm^2^ in their prospective study. Since Schegerer et al. [33] calculated mean radiation dose of TACE by sampling 16 hospitals from 13 countries in Europe in their prospective study, it seems to have reliability for the mean radiation dose of TACE. In comparison, the radiation of TACE was significantly lower in our study and this discrepancy is encouraging. However, there are several limitations because it is impossible to compare the detailed differences in our study with the TACE protocols of each hospital, such as whether CBCT was performed as a routine procedure and parameter differences such as the ratio of CBCT to total radiation dose. Jonczyk et al. [21] reported that the overall cumulative DAP was 318.9 ± 22.3 Gy cm^2^ and fluoroscopy time was 1020 ± 551 s. Furthermore, Jonczyk et al. [21] reported that even though CBCT adds radiation exposure in TACE, the capability of CBCT to detect vessels and overlay in real time during fluoroscopy helps to facilitate TACE with a resultant reduction in DAPs of up to 46%, which could result in the total radiation dose of adding CBCT to TACE being lower than TACE without adding CBCT. However, in daily practice, it is difficult not to perform DSA at all after the first CBCT acquisition, and there are advantages of additional acquisition of DSA. Thus, it seems difficult to reduce the radiation dose by 46% even if CBCT is utilized to its fullest. Even so, when CBCT is utilized to the maximum, it seems that the radiation dose reduction effect is enough to offset the additional 10 to 15% of radiation dose received from CBCT. More research is needed in this area.

We observed that the fluoroscopy time and DAP were significantly lower in group I than in group II. This finding was probably due to technical challenges in the case of multiple tumors and feeding arteries for selection, whereas single tumors and feeding arteries are relatively easier to treat.

The severity of liver dysfunction plays an important role in the management of HCC and is tightly linked with the long-term outcomes of patients who underwent TACE. In our study, we evaluated the liver function of patients before TACE using the Child–Pugh score. However, this score has limitations due to inter-related objective variables and subjective variables such as ascites and hepatic encephalopathy. Recently, noninvasive models such as the Model for End-stage Liver Disease (MELD), Albumin-Bilirubin (ALBI) grade, Easy (EZ)-ALBI grade, Platelet-Albumin-Bilirubin (PALBI), and Platelet-Albumin (PAL) have been used to evaluate liver function [34]. A recent study by Ho et al. [34] compared the prognostic performance of these noninvasive models for liver dysfunction and concluded that albumin-based liver reserve models are better prognostic tools than MELD score in HCC patients undergoing TACE. Of these, the PALBI score is the best model to evaluate the liver reserve. Future additional studies could consider using the PALBI score for pre-TACE evaluation.

In our study, the post-treatment radiological response was assessed using mRECIST at one month after TACE. Furthermore, we did not attach much significance to transient liver enzyme elevations that occurred after TACE, interpreting them all as PES. However, according to Granito et al. [35], post-TACE transient transaminase elevation was predictive of an objective response to superselective TACE in clinical practice. This is a very interesting result and, along with mRECIST, could serve as a useful clinical indicator for assessing the efficacy of TACE.

There are several limitations to the current study that should be considered when interpreting the results. First, this study was retrospectively designed, which may have introduced selection bias. Second, we made no direct comparison of the fluoroscopy time and radiation dose between TACE with DSA alone and TACE combined with CBCT because all procedures were routinely performed with CBCT guidance for detecting tumors and feeding arteries.

## 5. Conclusions

In conclusion, we found that when CBCT was added to the TACE procedure, TACE was safe, effective, and associated with acceptable complication and technical success rates. Furthermore, the addition of CBCT to DSA facilitated three-dimensional vessel navigation for superselection, which helped increase the detection rate of the tumors and feeding arteries without significantly increasing the radiation exposure.

## Figures and Tables

**Figure 1 medicina-59-01121-f001:**
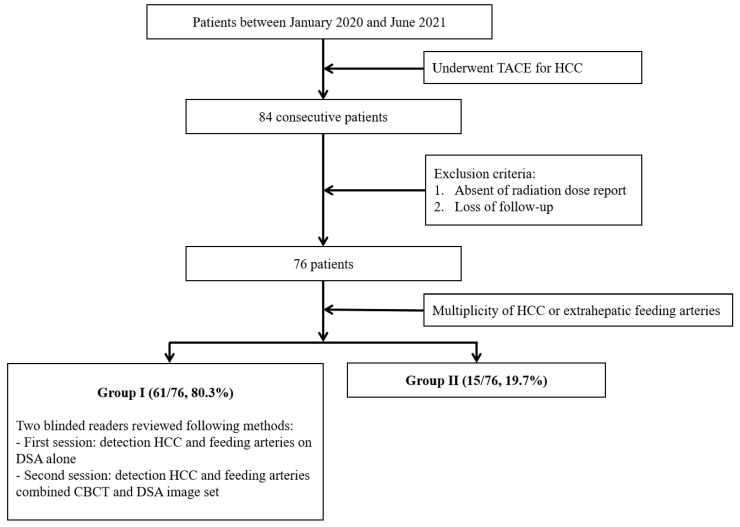
Flowchart of the case accrual process.

**Figure 2 medicina-59-01121-f002:**
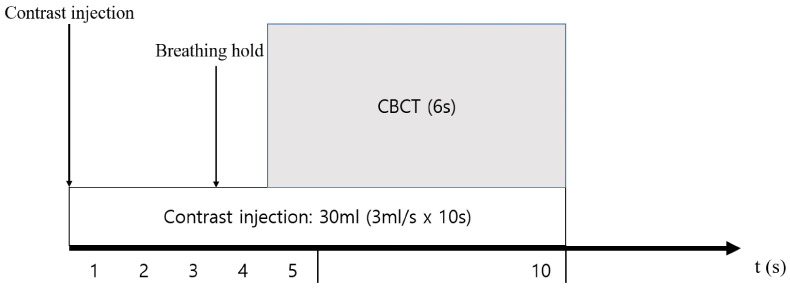
Routine single-phase cone-beam computed tomography (CBCT) protocol for hepatocellular carcinoma. Contrast media are infused for 10 s at 3 mL/s, and CBCT images are acquired after the patient holds their breath.

**Figure 3 medicina-59-01121-f003:**
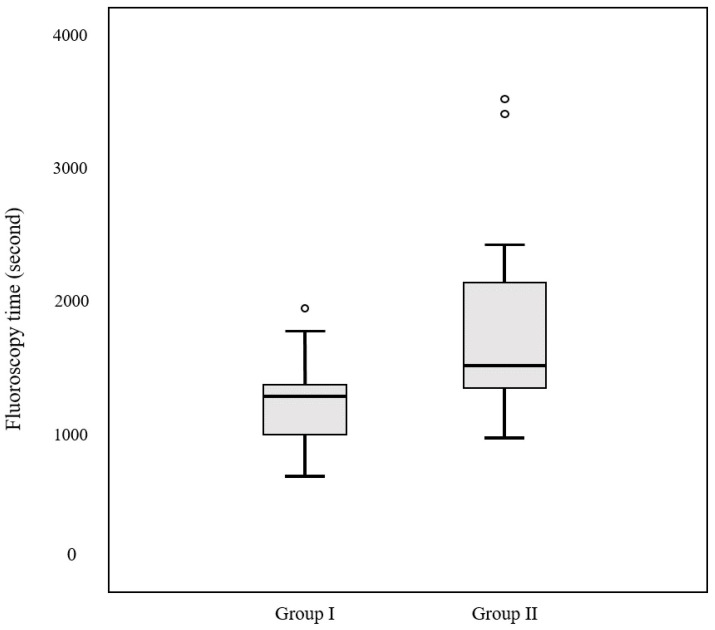
Box and whisker plot comparing the fluoroscopy time between groups I and II. The mean fluoroscopy time is 317 ± 213.6 s in group I and 153.6 ± 54.9 s in group II (*p* = 0.01).

**Figure 4 medicina-59-01121-f004:**
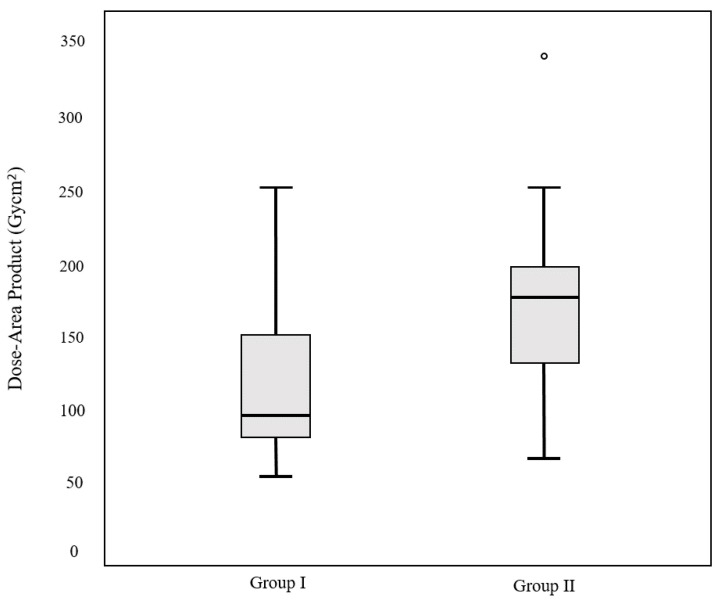
Box and whisker plot comparing the dose–area production (DAP) between groups I and II. The mean DAP is 7.9 ± 4.1 Gy cm^2^ in group I and 5.4 ± 1.9 Gy cm^2^ in group II (*p* = 0.016).

**Figure 5 medicina-59-01121-f005:**
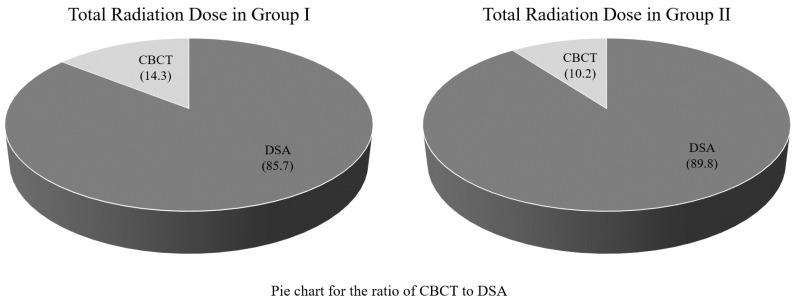
Pie chart showing the ratio of cone-beam computed tomography (CBCT) to digital subtraction angiography (DSA).

**Figure 6 medicina-59-01121-f006:**
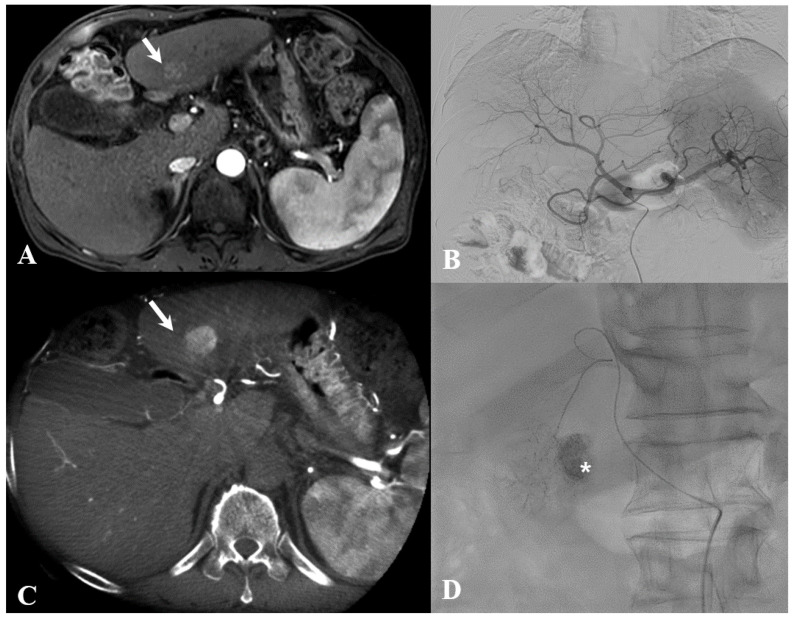
A 54-year-old man with a single hepatocellular carcinoma. (**A**) Arterial phase of a magnetic resonance imaging (MRI) scan demonstrates a hypervascular mass (white arrow) in segment 3 of the liver. (**B**) Arterial phase of celiac angiography does not show a small hypervascular tumor. (**C**). Cone-beam computed tomography (CBCT) increases sensitivity for detecting hypervascular tumor (white arrow), which is hard to see using digital subtraction angiography (DSA). (**D**) Successful transcatheter arterial chemoembolization (TACE) is performed after advanced applications (EmboGuide, Philips, Healthcare, Best, the Netherlands) providing three-dimensional vessel navigation for superselection. Note the treated lesion with good lipiodol uptake (asterisk).

**Table 1 medicina-59-01121-t001:** Procedural data reported for groups I and II and overall.

	Group I (*n* = 61)	Group II (*n* = 15)	Statistical Significance (*p*-Value)
Technical success (%)	100	100	
mRECIST (n, %)			
CR	38 (62.3)	4 (26.7)	*p* > 0.05
PR	17 (27.9)	8 (53.3)	
SD	1 (1.6)	1 (6.7)	
PD	5 (8.2)	2 (13.3)	
FT (s)	1366.2 ± 524.4	1822.7 ± 779.1	*p* < 0.05
DAP (Gy cm^2^)			
DSA alone	109.7 ± 63.2	156.2 ± 66.4	*p* < 0.05
CBCT	18.4 ± 7.4	17.8 ± 6.1	*p* < 0.05
DSA + CBCT	128.1 ± 66.7	174 ± 69.2	*p* < 0.05
Ratio (CBCT/Total)	14.3	10.2	
Complications (%)	2 (3.3)	1 (6.7)	*p* > 0.05

mRECIST, modified Response Evaluation Criteria in Solid Tumors; CR, complete response; PR, partial response, SD, stable disease; PD, progression of disease; FT, fluoroscopy time; DAP, dose–area product; CBCT, cone-beam computed tomography.

**Table 2 medicina-59-01121-t002:** Detection rate of hepatocellular carcinoma (HCC) and feeding arteries for digital subtraction angiography (DSA) and combined cone-beam computed tomography (CBCT) and DSA for both readers.

	DSA Alone	DSA + CBCT
Reader 1		
HCC, *n* (%)	78/112 (69.6)	109/112 (97.3)
Feeding arteries, *n* (%)	70/116 (60.3)	112/116 (96.6)
Reader 2		
HCC, *n* (%)	78/112 (69.6)	108/112 (96.4)
Feeding arteries, *n* (%)	74/116 (63.8)	113/116 (97.4)

## Data Availability

The dataset generated and/or analyzed during the current study is available from the corresponding author upon reasonable request.

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
