# Peer review of "Added Value of Cone-Beam Computed Tomography for Detecting Hepatocellular Carcinomas and Feeding Arteries during Transcatheter Arterial Chemoembolization Focusing on Radiation Exposure"

_medicina, 2023, doi:10.3390/medicina59061121_

Round 1
Reviewer 1 Report
The research article “Added value of cone-beam computed tomography for detecting hepatocellular carcinomas and feeding arteries during transcatheter arterial chemoembolization focusing on radiation exposure” by DJ Kim et al. emphasizes the benefit of using cone-beam computed tomography (CBCT) for detecting hepatocellular carcinomas (HCC) and feeding arteries during transcatheter arterial chemoembolization (TACE). The authors show that CBCT can increase sensitivity for detecting HCCs and feeding arteries without significantly increasing the radiation exposure.
The article is well-structured and findings are reported well and illustrated adequately. No major issues are to be reported.
Congratulations to the authors on this excellent work.
Author Response
Response to Reviewer 1 Comments
9 June, 2023
Dear reviewer 1 and editorial staffs in Medicina
We are sincerely grateful for your thorough consideration and scrutiny of our manuscript, “Added Value of Cone-beam Computed Tomography for Detecting Hepatocellular Carcinomas and Feeding Arteries During Transcatheter Arterial Chemoembolization Focusing on Radiation Exposure”, control number medicina-2420898. Through the accurate comments made by the reviewers, we better understand the critical issues in this paper. We hope that our revised manuscript will be considered and accepted for publication in the Medicina. We acknowledge that the scientific and clinical quality of our manuscript was improved by the scrutinizing efforts of the reviewers and editors.
Point 1: The research article “Added value of cone-beam computed tomography for detecting hepatocellular carcinomas and feeding arteries during transcatheter arterial chemoembolization focusing on radiation exposure” by DJ Kim et al. emphasizes the benefit of using cone-beam computed tomography (CBCT) for detecting hepatocellular carcinomas (HCC) and feeding arteries during transcatheter arterial chemoembolization (TACE). The authors show that CBCT can increase sensitivity for detecting HCCs and feeding arteries without significantly increasing the radiation exposure.
The article is well-structured and findings are reported well and illustrated adequately. No major issues are to be reported.
Congratulations to the authors on this excellent work.
Response 1: We are sincerely grateful for your thorough consideration and scrutiny of our manuscript.

Reviewer 2 Report
TITLE
In my opinion the title is interesting and attractive.
KEYWORDS
Authors did not correctly report all keywords from MeSH Browser. In particular, for example, I checked “transarterial chemoembolization,” on MeSH Browser and I did not find this KW. This is important, in my personal opinion, in order to increase the traceability of this paper (and consequently the possibility of the Journal to be cited by Readers and Stakeholders). I suggest to check all KWs and change those not reported on MeSH Browser.
ABSTRACT
The abstract is well structured and properly reflects the main text highlighting only the most important aspects of this paper, consequently no major adjustments are needed.
INTRODUCTION
In my opinion, the introduction could be improved by the Authors.
The authors could describe with greater attention some aspects of the main topics.
It is well known that one of the limitations of TACE is its standardization. Recently, a paper tried to standardize TACE [Ann Hepatol. 2021 May-Jun;22:100278. doi: 10.1016/j.aohep.2020.10.006. Epub 2020 Oct 29. PMID: 33129978.]. Please, could you discuss this theme according to the above-mentioned paper?
Materials and Methods _Patients
Please, could the authors explain more in detail how they defined “possible superselection of tumor/feeding arteries”?
Furthermore, the authors stated that: “At the first session, each reader reviewed the DSA images from patients in group I. Each reader recorded whether the tumor or feeding arteries were detected according to a 5-point confidence scoring system:”. Did they use a quantitative analysis? For example, arterial size? Toruosity?
Results section
It is clear.
In my opinion the authors could introduce some possible novelties such as [J Nucl Med 2022;63:556–559].
SUGGESTIONS
Could the authors discuss this future scenario: in Europe, some countries in Asia and USA, the surveillance program and also the future strategy to improve the surveillance program [Non-enhanced magnetic resonance imaging as a surveillance tool for hepatocellular carcinoma: Comparison with ultrasound. J Hepatol. 2020;72(4):718-724. doi:10.1016/j.jhep.2019.12.001 ----- Proposal of a new diagnostic algorithm for hepatocellular carcinoma based on the Japanese guidelines but adapted to the Western world for patients under surveillance for chronic liver disease. J Gastroenterol Hepatol. 2016;31(1):69-80. doi:10.1111/jgh.13150], will allow to overcome the ultrasound limitations in the detections of HCC at an early and very early stages.
In fact, actually ultrasound identify 4 out 10 patients in very early or early stages [Surveillance Imaging and Alpha Fetoprotein for Early Detection of Hepatocellular Carcinoma in Patients With Cirrhosis: A Meta-analysis. Gastroenterology. 2018;154(6):1706-1718.e1. doi:10.1053/j.gastro.2018.01.064]. The first consequence will be the detection of even-increasing number of lesions in very early and early stage (small lesions). Could the Authors discuss these themes, cite the papers and report their ideas about these possible scenarios? For example: will the treatment of choice change in the future if we will able to identify more lesions, even smaller? The routinely use of CBCT could help in the choice of TACE also for more small lesions?
A multitude of TACE-specific staging systems have been developed to use in pre-procedural phase, such as albumin-based liver reserve models [Cancers (Basel). 2023;15(7):1925. Published 2023 Mar 23. doi:10.3390/cancers15071925]. Furthermore, also postprocedural parameters have been investigated as outcome predictors of TACE’s efficacy, such as the Transient Hypertransaminasemia after TACE [J Pers Med. 2021;11(10):1041. Published 2021 Oct 17. doi:10.3390/jpm11101041].
Did the authors use these pre- and post-procedural evaluations? Please, may the authors mention it citing the above-mentioned papers?
REFERENCES
References reflect the style showed in the “Instruction for Authors”.
Author Response
Response to Reviewer 2 Comments
9 June, 2023
Dear reviewer 2 and editorial staffs in Medicina
We are sincerely grateful for your thorough consideration and scrutiny of our manuscript, “Added Value of Cone-beam Computed Tomography for Detecting Hepatocellular Carcinomas and Feeding Arteries During Transcatheter Arterial Chemoembolization Focusing on Radiation Exposure”, control number medicina-2420898. Through the accurate comments made by the reviewers, we better understand the critical issues in this paper. We have revised the manuscript according to the Reviewer’s suggestions. We hope that our revised manuscript will be considered and accepted for publication in the Medicina. We acknowledge that the scientific and clinical quality of our manuscript was improved by the scrutinizing efforts of the reviewers and editors.
The changes within the revised manuscript were highlighted (in red). Point-by-point responses to the reviewers’ comments are provided below.
Point 1: Authors did not correctly report all keywords from MeSH Browser. In particular, for example, I checked “transarterial chemoembolization,” on MeSH Browser and I did not find this KW. This is important, in my personal opinion, in order to increase the traceability of this paper (and consequently the possibility of the Journal to be cited by Readers and Stakeholders). I suggest to check all KWs and change those not reported on MeSH Browser.
Response 1: We appreciate reviewer’s comment. We have confirmed that “transarterial chemoembolization” or “transcatheter arterial chemoembolization” are not found on the MeSH Browser, and have replaced it with “chemoembolization”, which is searchable on the MeSH Browser. Additionally, to increase the traceability of this paper and to clarify the target lesion, we have added “hepatocellular carcinoma” in the keywords.
We have updated manuscript (keywords) as below.
Keywords: chemoembolization; hepatocellular carcinoma; cone-beam computed tomography; radiation exposure; sensitivity
Point 2: In my opinion, the introduction could be improved by the Authors. The authors could describe with greater attention some aspects of the main topics. It is well known that one of the limitations of TACE is its standardization. Recently, a paper tried to standardize TACE [Ann Hepatol. 2021 May-Jun;22:100278. doi: 10.1016/j.aohep.2020.10.006. Epub 2020 Oct 29. PMID: 33129978.]. Please, could you discuss this theme according to the above-mentioned paper?
Response 2: We appreciate the reviewer’s comment and find the paper you provided, [Ann Hepatol. 2021 May-Jun;22:100278. doi: 10.1016/j.aohep.2020.10.006. Epub 2020 Oct 29. PMID: 33129978.], to be highly interesting. We acknowledge that the paper you provided can plays a crucial role to set up for standardization of cTACE, and we briefly discuss the findings of referenced paper to enhance the introduction section.
We have updated manuscript (introduction) as below and the reference numbering to correspond with the newly added references.
Hepatocellular carcinoma (HCC) is the most common primary liver cancer accounting for 70% to 90% cases, and it is the sixth most commonly diagnosed and fourth most fatal cancer worldwide [1-3]. The Barcelona Clinic Liver Cancer staging system, which considers the tumor stage, liver function, and performance status, can be used to help recommend the treatment strategy according to each stage [4,5]. Transcatheter arterial chemoembolization (TACE) is the first-line treatment option for patients with the intermediate stage of HCC [6-9]. TACE can be classified into three types, conventional TACE (C-TACE), TACE with drug-eluting beads (DEB-TACE), and balloon occluded TACE (B-TACE) [7]. C-TACE was the first established option to treat the intermediate stage of HCC and works by delivering a cytotoxic chemotherapeutic agent saturated with ethiodized oil (Lipiodol, Guerbet, Villepinte) and occluding tumor-feeding arteries with an embolic agent to maximize cytotoxic and ischemic effects [6,7]. Although C-TACE has been reported to be associated with survival gain in patients with HCC, it has also been associated with a high incidence of systemic toxicity induced by the rapid circulation of chemotherapeutic agent-loaded lipiodol [10, 11]. Additionally, cTACE presents numerous challenges due to the absence of standardization at every phase [12]. This deficiency in standardization introduces a wide range of biases that hinder the comparison of previously published patient series who underwent cTACE treatment. In recent, Renzulli et al. [13] evaluated in vitro chemical and physical characteristics and behaviors over time of emulsions for cTACE prepared using two different methods and to assess the intra- and inter-operator variabilities in the preparation processes. In this interesting study, the mean droplet diameter decreased non-significantly when the number of pumping exchanges increased and increased significantly over time for both water-in-oil and chemotherapeutic-in-oil techniques. The droplets returned to their initial diameters after re-mixing and there were no significant differences in the intra- and inter-operator variabilities. According to their investigation, any interventional radiologists, regardless of their experience, may prepare these emulsions and these data may represent a set of instructions to standardize cTACE.
High incidence of systemic toxicity and lack of standardization of cTACE led to the development of DEB-TACE, which uses microspheres as drug carriers to slowly release chemotherapeutic agents and substantially reduce the concentration of systemic chemotherapeutic agents [6,14,15].
Reference.
- Marelli, L.; Stigliano, R.; Triantos, C.; Senzolo, M.; Cholongitas, E.; Davies, N.; Tibballs, J.; Meyer, T.; Patch, DW.; Burroughs, AK. Transarterial therapy for hepatocellular carcinoma: which technique is more effective? A systematic review of cohort and randomized studies. Cardiovasc Intervent Radiol 2007, 30:6–25, doi: 10.1007/s00270-006-0062-3.
- Renzulli, M.' Peta, G.; Vasuri, F.; Marasco, G.; Caretti, D.; Bartalena, L.; Spinelli, D.; Giampalma, E.; D'Errico, A.; Golfieri, R. Standardization of conventional chemoembolization for hepatocellular carcinoma. Ann Hepatol. 2021, 22, 100278. doi: 10.1016/j.aohep.2020.10.006.
Point 3: Please, could the authors explain more in detail how they defined “possible superselection of tumor/feeding arteries”? Furthermore, the authors stated that: “At the first session, each reader reviewed the DSA images from patients in group I. Each reader recorded whether the tumor or feeding arteries were detected according to a 5-point confidence scoring system:”. Did they use a quantitative analysis? For example, arterial size? Toruosity?
Response 3: We appreciate the reviewer’s comment. We will answer each of your questions separately.
- We divided the patients into Group I and II based on the feasibility of superselective TACE. This distinction was made because our study is focused on radiation dose, and we deemed the superselection of the lesion to be a significant factor affecting the radiation dose and fluoroscopy time. Therefore, the definition of "possible superselection of tumor/feeding arteries" in Group I is that hypervascular tumor(s) is/are clearly visible on CT or MRI, the number of such lesion(s) is/are relatively small (less than 3-4), and the feeding arteries are visible or expected to be clearly visible. In other words, group I is cases which superselective TACE is expected to be feasible. On the other hand, Group II was defined as cases which it was difficult or meaningless to perform superselective TACE due to multiple HCCs (5-6 or more), each having too many or hard to find feeding arteries, or cases of diffuse infiltrative HCC where the meaning of selection itself is absent, or cases with multiple extrahepatic feeding arteries making treatment difficult with a single TACE session.
- The detection of tumors and feeding arteries by each reader was a qualitative analysis based solely on whether the tumors and feeding arteries were visible on the DSA images and combined DSA with CBCT. We did not perform a quantitative analysis, and the reason for this is that an objective data needs to be provided for a quantitative analysis, which is realistically difficult in DSA images. For example, in the case of arterial size measurement, DSA images are obtained by projecting X-rays from one direction to get a 2D image, and the size of the artery can be measured differently depending on the distance between the object, the X-ray source, and the detector. Also, in the process of performing angiography, a temporary spasm may occur in the artery when passing through a hydrophilic wire, so the artery may appear well but is evaluated as small due to the spasm, increasing the false negative rate. On the other hand, CBCT has a set protocol and acquires a 3D image by rotating the gantry 180 degrees, which allows for obtaining objective data about the arterial size. Due to these differences, performing a quantitative analysis may greatly underestimate DSA, and there may be a possibility of objectively evaluating only CBCT. This could lead to a kind of measurement bias, where the performance of CBCT appears overwhelmingly superior to DSA.
Point 4: Could the authors discuss this future scenario: in Europe, some countries in Asia and USA, the surveillance program and also the future strategy to improve the surveillance program [Non-enhanced magnetic resonance imaging as a surveillance tool for hepatocellular carcinoma: Comparison with ultrasound. J Hepatol. 2020;72(4):718-724. doi:10.1016/j.jhep.2019.12.001 ----- Proposal of a new diagnostic algorithm for hepatocellular carcinoma based on the Japanese guidelines but adapted to the Western world for patients under surveillance for chronic liver disease. J Gastroenterol Hepatol. 2016;31(1):69-80. doi:10.1111/jgh.13150], will allow to overcome the ultrasound limitations in the detections of HCC at an early and very early stages.
In fact, actually ultrasound identify 4 out 10 patients in very early or early stages [Surveillance Imaging and Alpha Fetoprotein for Early Detection of Hepatocellular Carcinoma in Patients with Cirrhosis: A Meta-analysis. Gastroenterology. 2018;154(6):1706-1718.e1. doi:10.1053/j.gastro.2018.01.064]. The first consequence will be the detection of even-increasing number of lesions in very early and early stage (small lesions). Could the Authors discuss these themes, cite the papers, and report their ideas about these possible scenarios? For example: will the treatment of choice change in the future if we will be able to identify more lesions, even smaller? The routinely use of CBCT could help in the choice of TACE also for more small lesions?
Response 4: We appreciate the reviewer’s comment. The various models and strategies you've provided for HCC surveillance are extremely valuable, and we've decided to actively incorporate them. We've updated our manuscript to include a discussion on recent models and studies related to HCC surveillance, as well as diagnostic algorithms, referencing the papers you recommended. In our study, the sensitivity of tumor and feeding arteries detection significantly increased when CBCT was added to DSA. Therefore, this suggests that CBCT could be considered as part of future strategies for HCC surveillance when HCC cannot be confirmed based on imaging alone. However, since our study was not intended for surveillance, we mentioned this as a limitation and modified the manuscript to indicate that more research will be needed in the future.
We have updated manuscript (discussion) as below and the reference numbering to correspond with the newly added references.
We found that the sensitivity for detecting HCC and feeding arteries significantly increased by approximately 20-35% after adding CBCT to DSA. In addition, the mean confidence level of both readers for detecting HCC and feeding arteries on CBCT was significantly higher than that on DSA alone. These results are concordant with previous studies. A recent systematic review and meta-analysis [27] showed that the pooled sensitivity of CBCT for detecting tumors and feeding arteries was 90% and 93%, respectively. In contrast, the pooled sensitivity of DSA for tumor detection and feeding arteries was 67% and 55%, respectively. Lucatelli et al. [28] also reported a similar result, detailing that dual-phase CBCT had a better diagnostic performance than pre-examination CT or MR in detecting and characterizing HCC during TACE with high sensitivity and accuracy. Therefore, adding CBCT during TACE procedure is definitely helpful in finding the tumor and its feeding vessel.
The greatly increased detection rate of tumors and feeding arteries when adding CBCT to DSA is quite encouraging. With recent advancements in imaging techniques, surveillance programs for HCC detection in chronic liver disease are being implemented in Europe, some Asian countries, and the United States, and future strategies to improve these surveillance programs are being proposed. According to a meta-analysis by Tzartzeva et al. [29] while USG alone has 84% sensitivity for HCC of any stage, it only has 45% sensitivity for early-stage HCC, detecting only 4 out of 10 patients. Their meta-analysis confirmed that the sensitivity of HCC detection increases to 63% when USG is combined with AFP measurement, and they recommend adding AFP to USG for surveillance in clinical practice. Furthermore, Park et al. [30] reported that USG showed 27.9% sensitivity and non-enhanced MRI showed 79.1% sensitivity for HCC surveillance in high-risk patients, suggesting that non-enhanced MRI could be a replacement modality for USG. A recent meta-analysis by Lee et al. [31] found that MRI had higher per-lesion sensitivity than CT (80% vs 68%, P = 0.0023) for evaluating HCC. Moreover, ethoxybenzyl-diethylenetriamine pentaacetic acid (EOB)-MRI showed significantly higher per-lesion sensitivity than MRI performed with other contrast agents (87% vs 74%, P = 0.03) [31]. Accordingly, EOB-MRI should be the preferred imaging modality for the diagnosis of HCC in patients with chronic liver disease. Based on these results, a new diagnostic algorithm for hepatocellular carcinoma using EOB-MRI, based on Japanese guidelines but adapted to the Western world for patients under surveillance for chronic liver disease, has been proposed in Europe [32]. In contrast to the sensitivity of 27.9~45% for USG alone, 63% USG with AFP, 46% for CT, 80% for MRI, and 80% for EOB-MRI shown in literature reviews [29-32], in our study, the sensitivity exceeded 95% when DSA and CBCT were interpreted together. This suggests that CBCT could be considered as part of future strategies for HCC surveillance when HCC cannot be confirmed based on imaging alone. However, further studies are needed to determine whether DSA, as an invasive technique, is superior to other invasive methods such as biopsy or noninvasive imaging modalities. Furthermore, our study may have a selection bias as we performed CBCT only on patients who were already diagnosed with HCC through surveillance. Further research, such as randomized clinical trials, is needed to address this issue.
However, the acquisition of CBCT inevitably results in additional radiation exposure, despite the benefit of adding CBCT to TACE. The mean dose of radiation of added CBCT was 18.3 ± 7.1 Gy cm², and the mean ratio of DAP of CBCT to total DAP was 13.3%. Compared with a recent study in which the results showed that additional CBCT increased DAP by 6% [21], our additional radiation dose of CBCT seems to be high. However, in their study, the total mean DAP was 318.9 Gy cm², which was 2.3 times higher than ours. Furthermore, the overall mean DAP of CBCT was 17.9 Gy cm² in their study, which was similar to the current study. Therefore, a ratio of 10 to 15% of CBCT to the total radiation dose in our study seems to be acceptable. Moreover, in our study, the total radiation dose of group II was significantly higher than group I, whereas the ratio of the radiation dose of CBCT to the total radiation dose was lower in group II than in group I. This suggests that as the total radiation dose increases, the ratio of the radiation dose of CBCT to the total radiation dose decreases. Based on these findings, even if the ratio of the radiation dose of CBCT to the total radiation dose increases, the total radiation dose would decrease, so this also seems acceptable.
References
- Tzartzeva, K.; Obi, J.; Rich, NE.; Parikh, ND.; Marrero, JA.; Yopp, A.; Waljee, A.; Singal, AG. Surveillance Imaging and Alpha Fetoprotein for Early Detection of Hepatocellular Carcinoma in Patients with Cirrhosis: A meta-analysis. Gastroenterology 2018, 154, 1706–1718.e1. doi: 10.1053/j.gastro.2018.01.064.
- Park, HJ.; Jang, HY.; Kim, SY.; Lee, SJ.; Won, HJ.; Byun, JH.; Choi, SH.; Lee, SS.; An, J.; Lim, YS. Non-enhanced Magnetic Resonance Imaging as a Surveillance Tool for Hepatocellular Carcinoma: Comparison with Ultrasound. J Hepatol. 2020, 72, 718–724, doi: 10.1016/j.jhep.2019.12.001.
- Lee, YJ.; Lee, JM.; Lee, JS.; Lee, HY.; Park, BH.; Kim YH.; Han, JK.; Choi, BI. Hepatocellular Carcinoma: Diagnostic Performance of Multidetector CT and MR imaging-a systematic review and meta-analysis. Radiology 2015, 275, 97–109, doi: 10.1148/radiol.14140690.
- Renzulli, M.; Golfieri, R. Proposal of a New Diagnostic Algorithm for Hepatocellular Carcinoma based on the Japanese Guidelines but Adapted to the Western World for Patients under Surveillance for Chronic Liver Disease. J Gastroenterol Hepatol. 2016, 31, 69–80, doi: 10.1111/jgh.13150.
Point 5: A multitude of TACE-specific staging systems have been developed to use in pre-procedural phase, such as albumin-based liver reserve models [Cancers (Basel). 2023;15(7):1925. Published 2023 Mar 23. doi:10.3390/cancers15071925]. Furthermore, also postprocedural parameters have been investigated as outcome predictors of TACE’s efficacy, such as the Transient Hypertransaminasemia after TACE [J Pers Med. 2021;11(10):1041. Published 2021 Oct 17. doi:10.3390/jpm11101041].
Did the authors use these pre- and post-procedural evaluations? Please, may the authors mention it citing the above-mentioned papers?
Response 4: We appreciate the reviewer’s comment. The articles you provided are very interesting and attractive, which capable of further improving our research scientifically. We will refer to and cite the papers you provided to supplement our discussion.
We have updated manuscript (discussion) as below and the reference numbering to correspond with the newly added references.
We observed that the fluoroscopy time and DAP were significantly lower in group I than in group II. This finding was probably due to technical challenges in the case of multiple tumors and feeding arteries for selection, whereas single tumors and feeding arteries are relatively easier to treat.
The severity of liver dysfunction plays an important role in the management of HCC and is tightly linked with long-term outcome of patients who underwent TACE. In our study, we evaluated the liver function of patients before TACE using the Child-Pugh score. However, this score has limitation due to inter-related objective variables and subjective variables such as ascites and hepatic encephalopathy. Recently, noninvasive models such as the Model for End-stage Liver Disease (MELD), Albumin-Bilirubin (ALBI) grade and Easy (EZ)-ALBI grade, Platelet-Albumin-Bilirubin (PALBI) and Platelet-Albumin (PAL) have been used to evaluate liver function [34]. A recent study by Ho et al. [34] compared the prognostic performance of these noninvasive models for liver dysfunction and concluded that albumin-based liver reserve models are better prognostic tools than MELD score in HCC patients undergoing TACE. Of these, the PALBI score is the best model to evaluate the liver reserve. Future additional studies could consider using the PALBI score for pre-cTACE evaluation
In our study, the post-treatment radiological response was assessed using mRECIST at one month after TACE. Furthermore, we did not attach much significance to transient liver enzyme elevations that occurred after TACE, interpreting them all as post-embolization syndrome (PES). However, according to Granito et al. [35], post-cTACE transient transaminase elevation was predictive of an objective response to superselective cTACE in clinical practice. This is a very interesting result and, along with mRECIST, could serve as a useful clinical indicator for assessing the efficacy of TACE.
There are several limitations to the current study that should be considered when interpreting the results. First, this study was retrospectively designed, which may have introduced selection bias. Second, we made no direct comparison of the fluoroscopy time and radiation dose between TACE with DSA alone and TACE combined with CBCT because all procedures were routinely performed with CBCT guidance for detecting tumors and feeding arteries.
References
- Ho, SY.; Liu, PH.; Hsu, CY.; Huang, YH.; Liao, JI.; Su, CW.; Hou, MC.; Huo, TI. Comparison of Four Albumin-Based Liver Reserve Models (ALBI/EZ-ALBI/PALBI/PAL) against MELD for Patients with Hepatocellular Carcinoma Undergoing Transarterial Chemoembolization. Cancers (Basel), 2023, 15, 1925, https://doi.org/10.3390/cancers15071925.
- Granito, A.; Facciorusso, A.; Sacco, R.; Bartalena, L.; Mosconi, C.; Cea, UV.; Cappelli, A.; Antonino, M.; Modestino, F.; Brandi, N.; Tovoli, F.; Piscaglia, F.; Golfieri, R.; Renzulli, M. TRANS-TACE: Prognostic Role of the Transient Hypertransaminasemia after Conventional Chemoembolization for Hepatocellular Carcinoma. J Pers Med. 2021, 11, 1041, doi: 10.3390/jpm11101041.
